# Tick Diversity and Pathogen Transmission in Daejeon, Korea: Implications from Companion Animals and Walking Trails

**DOI:** 10.3390/vetsci11020090

**Published:** 2024-02-14

**Authors:** Jinwoo Seo, Gyurae Kim, Jeong-ah Lim, Seungho Song, Dae-Sung Yoo, Ho-Seong Cho, Yeonsu Oh

**Affiliations:** 1Division of Animal Health, Daejeon Institute of Health and Environment, Daejeon 34142, Republic of Korea; jinu0205@korea.kr (J.S.); dalpongi@korea.kr (J.-a.L.); ssh6370@korea.kr (S.S.); 2College of Veterinary Medicine and Institute of Veterinary Science, Kangwon National University, Chuncheon 24341, Republic of Korea; twin0852@kangwon.ac.kr; 3College of Veterinary Medicine, Chonnam National University, Gwangju 61186, Republic of Korea; shanuar@chonnam.ac.kr; 4College of Veterinary Medicine and Bio-Safety Research Institute, Jeonbuk National University, Iksan 54596, Republic of Korea; hscho@jbnu.ac.kr

**Keywords:** tick-borne diseases (TBDs), climate change, global warming, companion animals, vector-borne diseases

## Abstract

**Simple Summary:**

Tick-borne diseases are increasingly spreading due to global warming. Although the migratory abilities of ticks are limited, the spread of ticks between companion animals and humans has increased with the recent increase in the number of companion animals. For this reason, it is important to know the species of ticks that are parasitizing companion animals and whether they carry disease-causing organisms. Our research team collected ticks from pet trails and pet hospitals in the Daejeon area over a period of three years and confirmed the presence of pathogens through polymerase chain reaction (PCR) analysis. As a result of examining approximately 29,200 ticks, the dominant species was found to be *H. longicornis*, and the presence of four types of pathogens was confirmed. Through this study, we confirmed the possibility of tick-borne disease transmission between companion animals and humans and the emergence of a new tick (*A. testudinarium*) due to global warming in the Daejeon area.

**Abstract:**

With the ongoing global warming-induced climate change, there has been a surge in vector-borne diseases, particularly tick-borne diseases (TBDs). As the population of companion animals grows, there is growing concern from a One Health perspective about the potential for these animals to spread TBDs. In this study, ticks were collected from companion animals and the surrounding environment in Daejeon Metropolitan City, Korea, using flagging and dragging, and CO_2_ trap methods. These ticks were then subjected to conventional (nested) PCR for severe fever with thrombocytopenia syndrome virus (SFTSV), *Anaplasma* spp., *Ehrlichia* spp., and *Borrelia* spp. We identified a total of 29,176 ticks, consisting of three genera and four species: *H. longicornis, H. flava, I. nipponensis,* and *A. testudinarium.* Notably, *H. longicornis* was the predominant species. The presence of *A. testudinarium* suggested that the species traditionally found in southern regions are migrating northward, likely as a result of climate change. Our PCR results confirmed the presence of all four pathogens in both companion animals and the surrounding environment, underscoring the potential for the indirect transmission of tick-borne pathogens to humans through companion animals. These findings emphasize the importance of the ongoing surveillance of companion animals in the management and control of TBDs.

## 1. Introduction

Vector-borne diseases (VBDs) encompass a group of illnesses initiated by pathogens that are transmitted through vectors such as arthropods. Mosquitoes and ticks are the leading transmitters of these diseases, with ticks being recognized as the second most significant vector globally, next to mosquitoes [1]. Ticks harbor a diversity of pathogenic agents including bacteria, viruses, and protozoa, collectively responsible for a plethora of tick-borne diseases (TBDs) that represent a significant threat to public health and animal welfare [2]. Notably, genera such as *Rickettsia*, *Anaplasma*, *Bartonella*, *Borrelia*, *Coxiella*, and *Ehrlichia* are among the medically important pathogens transmitted by ticks [2,3]. 

The progression of global warming has exacerbated various environmental problems, notably the surge in VBDs [4]. Ticks, being ectothermic organisms with extensive outdoor lifecycles, are particularly susceptible to environmental changes. Humidity levels above 85%, coupled with temperatures over 7 °C, create ideal conditions for their reproductive and developmental processes, subsequently heightening their activity and the pace of disease transmission [5]. These conditions, which are becoming more prevalent due to climate change, enhance the ticks’ lifecycle and disease transmission rates, broadening their geographic spread [6]. 

Canada serves as a prime example of this trend. Before the 1990s, tick-borne infections in humans were rare, with diseases like Powassan encephalitis, tularaemia, and Rocky Mountain spotted fever being exceptions [7]. This changed when the tick species *I. scapularis* extended its range from the northeastern United States into southeastern Canada, introducing Lyme borreliosis to the area, which has now become endemic [8,9]. Similar patterns of tick migration and disease prevalence have been observed in European countries, including Scandinavia [10,11].

Moreover, the rise in pet ownership is drawing increased attention to TBDs [12]. Most of these diseases are zoonotic, meaning they can transfer between animals and humans. Studies show that pet owners have higher likelihood of encountering ticks, and therefore a greater risk of TBDs [13]. This correlation emphasizes the importance of a One Health approach, considering the interconnected health of people, animals, and the environment.

In Korea, traditionally characterized by a temperate climate with four distinct seasons, there is a noticeable shift towards a subtropical climate due to global warming. This has led to shorter winters and extended summers, with predictions indicating a significant alteration in seasonal lengths by the latter half of the century. Such climatic changes are creating more favorable conditions for ticks and, consequently, an increased risk of TBDs [14,15,16].

Korea is home to 28 tick species across six genera, with the most prevalent being *Haemaphysalis longicornis*, *Haemaphysalis flava*, and *Ixodes nipponensis*. Among these, *H. longicornis* is the most common, accounting for over 78% of tick populations. Notable diseases in Korea include severe fever with thrombocytopenia syndrome (SFTS), Lyme disease (*Borrelia* spp.), spotted fever group (SFG) rickettsioses, ehrlichiosis and anaplasmosis, Q fever (*Coxiella burnetii*), and *Bartonella* spp. infections [17]. SFTS, in particular, is of concern due to its high mortality rate and its increasing incidence since its first report in Korea in 2013 [18]. 

Given these challenges, the importance of monitoring tick populations, especially on pets and in environments these animals frequent, cannot be overstated. Despite ongoing surveys in the general environment, the study of ticks on companion animals has been less thorough. Addressing this gap is crucial for the development of effective control measures against TBDs. This study aims to investigate the distribution of ticks in areas commonly visited by pets and to analyze the prevalence of major pathogens such as SFTSV, *Anaplasma* spp., *Ehrlichia* spp., and *Borrelia* spp. This research is pivotal for developing comprehensive control strategies against TBDs, particularly in the face of evolving environmental societal dynamics.

## 2. Materials and Methods

### 2.1. Sample Collection

The collection of ticks was conducted across various locations, including urban walking paths and trails frequented by companion animals, and directly from the companion animals themselves. Specifically, within the urban areas of Daejeon Metropolitan City, Korea, our team selected 15 distinct sites distributed over five districts. This selection process involved identifying three trails close to residential zones within each district (Table 1, Figure 1). 

Throughout the year 2019, ticks were systemically gathered monthly from these sites, as detailed in Table 1. The initial phase of collection was performed using the flagging and dragging technique. This entailed sweeping a cloth measuring 1 m by 1.2 m over an area of approximately 100 square meters. Subsequently, to enhance the efficacy of our collection, we deployed a CO_2_ baiting trap, utilizing dry ice, at the same location. Ticks attracted to the trap were then collected on the following day. 

For the trails used by companion animals, our selection targeted 10 locations within Daejeon Metropolitan City, Korea. These were evenly distributed, with two trails chosen from each of the five districts. In 2020, monthly tick collections at these sites were carried out employing a CO_2_ baiting trap that utilized dry ice.

Regarding the collection of ticks from companion animals, we sourced specimens from pets that were brought to veterinary clinics as well as from abandoned animals housed at animal shelters in Daejeon Metropolitan City from 2021 to 2022. To ensure the safe and efficient removal of ticks without causing harm to the animal or the tick itself, we provided veterinary hospitals and animal shelters with a specialized tick removal tool known as the Tick Key. 

Once collected, the ticks were placed into tick collection tubes (SPL Life Sciences Co., Pocheon, Republic of Korea) and were immediately stored at an ultra-low temperature of −70 °C in a deep freezer for preservation.

### 2.2. Identification of Tick Species and Developmental Stages

The ticks gathered during our study were subjected to meticulous examination to determine their species and developmental stages. Using an Olympus SZ61 (Olympus Co., Tokyo, Japan), we assessed a range of morphological features. These included the lengths of the idiosoma and gnathosoma, the enamel pattern on the scutum, the presence of festoons, the configuration of the genital aperture, the lengths of the palps and palp articles, as well as the coloration of the ticks’ torsos and legs [19].

After conducting a morphological analysis, all tick samples underwent molecular studies. DNA extraction was performed using the QIAamp Blood and Tissue Kit (Qiagen, Hilden, Germany), following the manufacturer’s protocol. The extracted DNA was then stored at −20 °C until further use. For each tick species, individual specimens were subjected to PCR to amplifying regions of the *cox1* and *16S* rRNA genes [20]. The PCR products were loaded onto 1.5% agarose gel in volumes of five microliters per sample, and the results were visualized using a gel documentation system. The PCR products were purified using the QIAquick^®^ PCR Purification Kit (Qiagen, Hilden, Germany), adhering to the manufacturer’s instructions, and subsequently sent for sequencing to Macrogen Company (Seoul, Republic of Korea). 

The sequences of the *cox1* and *16S* rRNA genes were manually edited and assembled, with bidirectional consensus sequences generated using BioEdit software version 7.2.6.1 [21]. These sequences were then compared to those available in the GenBank database using the Basic Local Alignment Search Tool (BLAST) on the National Center for Biotechnology Information (NCBI) website, available at “https://blast.ncbi.nlm.nih.gov/Blast.cgi (accessed on 30 September 2023)”.

### 2.3. DNA Extraction for Polymerase Chain Reaction Amplification

The ticks collected were assorted into pools based on their developmental stage, species, and site of collection in order to streamline the DNA extraction process for PCR analysis (with up to 50 larvae, 30 nymphs, and 10 adults per pool). Each pooled sample of ticks was placed into a grinding tube, which also contained 2.8 mm stainless-steel beads and 400 μL of sterile phosphate-buffered saline (PBS). These samples were then subjected to mechanical homogenization using a Precellys 24 tissue homogenizer (Bertin Technologies Co., Bretonneux, France), operating at 6000 rpm in 30 s intervals. This process was repeated three times to ensure thorough disruption of the tick tissue. 

Following homogenization, the tick mixtures were centrifuged at a high speed of 12,000 rpm for 10 min at a temperature of 4 °C to sediment the solid fragments and obtain a clear supernatant. The DNA and RNA were extracted from the homogenate utilizing the TANBead Nucleic Acid Extraction Kit (TANBead, Taoyuan, Taiwan) according to the manufacturer’s instructions.

### 2.4. Real-Time Polymerase Chain Reaction

In order to accurately detect pathogens, our research team first performed real-time PCR and then additionally performed conventional (nested) PCR. All PCR processes were performed using the CFX96 Touch Real-time PCR System (Bio-Rad, Hercules, CA, USA). To detect SFTSV and *Ehrlichia*/*Anaplasma* spp., the PowerChek SFTS (S/M segment) Real-time PCR kit (Kogenebiotech, Seoul, Republic of Korea) and the PowerChek *Ehrlichia*/*Anaplasma* Real-time PCR kit (Kogenebiotech, Republic of Korea) were used according to the manufacturer’s instructions. To detect *Borrelia* spp., real-time PCR was performed using primers B1 and B2 to target *16S* rRNA (Table 2). The PCR assay was conducted in 20 μL reaction mixtures with 10 μL of 2× TOPsimple DyeMix (aliquot)-HOT (Enzynomics, Daejeon, Republic of Korea), 1 μL of each primer (10 pmol/μL), 1 μL of template DNA, and 7 μL of DW. The reaction conditions comprised a denaturation step at 95 °C for 10 min. Following this, 35 cycles of 95 °C for 30 s, 52 °C for 30 s, and 72 °C for 45 s were performed, followed by a final extension at 72 °C for 5 min.

### 2.5. SFTSV Conventional (Nested) Polymerase Chain Reaction

To detect SFTSV, nested PCR was performed using MF3 and MR2 primers to target the M fragment (Table 2). Reactions were performed in a total volume of 20 μL, containing TOPscript One-step RT PCR DryMix (Enzynomics, Republic of Korea), 1 μL of each primer (10 pmol/μL), 5 μL of template RNA, and 13 μL of distilled water (DW). The reaction conditions comprised cDNA synthesis steps at 50 °C for 30 min and at 95 °C for 10 min. Following this, 35 cycles of 95 °C for 20 s, 58 °C for 40 s, and 72 °C for 30 s were performed, followed by a final extension at 72 °C for 5 min. For the second amplification, reactions were performed in a total volume of 20 μL, containing 10 μL of 2× TOPsimple DyeMix (aliquot)-HOT (Enzynomics, Republic of Korea), 1 μL of each primer (10 pmol/μL), 1 μL of ^1st^PCR product, and 7 μL of DW, using MMF3 and MMF2 primers. Reaction conditions comprised a denaturation step at 94 °C for 5 min, followed by 35 cycles of 94 °C for 20 s, 59 °C for 20 s, and 72 °C for 20 s. After this, a final extension was performed at 72 °C for 5 min.

### 2.6. Anaplasma/Ehrlichia Detection Using Conventional (Nested) Polymerase Chain Reaction

To detect *Anaplasma* spp. and *Ehrlichia* spp., nested PCR was performed by using primers AE1-F1 and AE1-R1 to target *16S rRNA* (Table 2). The reactions were performed in a total volume of 20 μL, containing TOPscript One-step RT PCR DryMix (Enzynomics, Republic of Korea), 1 μL of each primer (10 pmol/μL), 1 μL of template RNA, and 7 μL of DW. The reaction conditions comprised a denaturation step at 95 °C for 10 min. Following this, 35 cycles of 95 °C for 30 s, 59 °C for 1 min, and 72 °C for 2 min were performed, followed by a final extension at 72 °C for 5 min. For the second amplification, the primers AP-F and AP-R were used for *Anaplasma* spp., and the primers EC-F2 and EC-R2 were used for *Ehrlichia* spp. The reactions were performed in a total volume of 20 μL, containing 10 μL of 2× TOPsimple DyeMix (aliquot)-HOT (Enzynomics, Republic of Korea), 1 μL of each primer (10 pmol/μL), 1 μL of ^1st^PCR product, and 7 μL of DW. The reaction conditions comprised a denaturation step at 95 °C for 10 min, followed by 35 cycles of 95 °C for 30 s, 56 °C for 30 s, and 72 °C for 45 s. After this, a final extension was performed at 72 °C for 5 min.

The minimum infection rate (MIR) of each pathogen was calculated in regard to collection location (urban walking trails, pet trails, and companion animals). The MIR was calculated by dividing the number of infected pools by the total number of ticks in all of the pools and expressed as the number of infected pools per 100 ticks tested, assuming that each positive pool only had one positive tick [25].

## 3. Results

### 3.1. Tick Species and Developmental Stages of Collected Ticks

Following morphological identification, molecular analysis of the *cox1* and *16S* rRNA genes confirmed the identification of all ticks as belonging to the following three genera and four species: *H. longicornis* (n = 25,936), *H. flava* (n = 2211), *I. nipponensis* (n = 1027), and *A. testudinarium* (n = 2). The most prevalent species were *H. longicornis*, followed by *H. flava*, *I. nipponensis*, and *A. testudinarium*. In terms of development stages, larvae were the most frequently collected (n = 13,296), followed by nymphs (n = 12,224) and adults (n = 3656).

### 3.2. Distribution of Tick and Pathogen Detection on Urban Walking Trails

A total of 16,803 ticks, including four species from three genera (14,949 *H. longicornis*, 1011 *H. flava*, 842 *I. nipponensis*, and 1 *A. testudinarium*) were collected from urban walking trails. Among all ticks, *H. longicornis* (89%) was the most commonly collected species, and irrespective of species, most of the ticks were collected in the larval stage.

In the PCR assay, SFTSV was detected in two pools of *H. longicornis*. Borrelia, the causative agent of Lyme disease, was detected in a total of thirty-two pools, including thirteen pools of *H. longicornis* (six adult pools, six nymph pools, and one larva pool), one pool of *H. flava*, and eighteen pools of *I. nipponensis* (fourteen adult pools and four nymph pools). The MIR of each pathogen was 0.01% for SFTSV and 0.19% for *Borrelia* spp. (Table 3).

### 3.3. Distribution of Ticks and Pathogen Detection on Pet Trails

A total of 11,016 ticks, including four species from three genera (9913 *H. longicornis*, 1059 *H. flava*, 43 *I. nipponensis*, and 1 *A. testudinarium*) were collected from pet trails. Among all ticks, *H. longicornis* (90%) was the most commonly collected species, and irrespective of species, most of the ticks were collected in the nymph stage.

In the PCR assay for ticks found on pet trails, SFTSV was detected in one pool of *H. longicornis* (one adult pool). *Borrelia* spp. was detected in three pools of *I. nipponensis* (three adult pools). Unlike on urban walking trails, *Ehrlichia* spp. was detected in a total of eight samples, including five pools of *H. longicornis* (three adult pools, two nymph pools), two pools of *H. flava* (one adult pool and one nymph pool), and one pool of *I. nipponensis* (one adult pool). The MIR of each pathogen was 0.01% for SFTSV, 0.07% for *Ehrlichia* spp., and 0.03% for *Borrelia* spp. (Table 4).

### 3.4. Distribution of Ticks and Pathogen Detection from Companion Animals

In 2021, a total of 346 ticks, including three species from two genera (310 *H. longicornis*, 1 *H. flava*, and 35 *I. nipponensis*), were collected from companion animals. Among all ticks, *H. longicornis* (90%) was the most commonly collected species, and irrespective of species, most of the ticks were collected in the adult stage. In the PCR assay, SFTSV was detected in one pool of *H. longicornis* (one adult pool). In the case of *Borrelia* spp., it was detected in one pool of *H. longicornis* (one adult pool). *Ehrlichia* spp. were detected in a total of three pools, including two pools of *H. longicornis* (two adult pools), and one pool of *I. nipponensis* (one adult pool). Unlike in other sites, *Anaplasma* spp. was detected in a total of three pools, including one pool of *H. longicornis* and two pools of *I. nipponensis*. The MIR of each pathogen was 0.29% for SFTSV, 0.87% for *Anaplasma* spp., 0.87% for *Ehrlichia* spp., and 0.29% for *Borrelia* spp. (Table 5).

In 2022, a total of 1011 ticks, including three species from two genera (764 *H. longicornis*, 140 *H. flava*, and 107 *I. nipponensis*), were collected. Among all ticks, *H. longicornis* (76%) was the most commonly collected species, and irrespective of species, most of the ticks were collected in the nymph stage. In the PCR assay, SFTSV was not detected. *Anaplasma* spp. was detected in a total of two pools, including one pool of *H. longicornis* (one adult pool) and one pool of *I. nipponensis* (one adult pool). *Borrelia* spp. was detected in one pool of *H. longicornis* (one adult pool). The MIR of each pathogen was 0.2% for *Anaplasma* spp. and 0.1% for *Borrelia* spp. (Table 6).

## 4. Discussion

This research concentrated on assessing the prevalence of ticks, both in areas within Daejeon Metropolitan City, Korea, frequently visited by pets and on the pets themselves. It also focused on integrating the exploration of four distinct pathogens. A substantial collection of ticks, totaling 27,819, was amassed for this study; 27,819 ticks were found on trails frequented by pets and 1357 were collected directly from pets. These specimens underwent detailed molecular analysis.

In categorizing the collected ticks, we identified a variety of species within the following three genera: *H. longicornis*, *H. flava*, *I. nipponensis*, and *A. testudinarium*. Notably, *H. longicornis* emerged as the predominant species across urban walking paths, along trails used by pets, and on the pets themselves. This distribution pattern closely mirrors those observed in areas where tick surveillance is regularly conducted [23,26]. Given that *H. longicornis* is a known primary vector for SFTSV, its widespread presence raises concerns about the potential for SFTSV to become a significant health issue in Korea. *H. longicornis* is also recognized as potentially transmitting other pathogens like *Borrelia* spp., *Anaplasma* spp., and *Ehrlichia* spp. [27,28,29]. Our detection of these four pathogens lends credence to their potential role as vectors. Additionally, *A. testudinarium*, typically found in the southern region of Korea (i.e., in Jeolla-do and Gyeongsang-do), has been detected in Daejeon, suggesting a northward expansion of its habitat [30].

Our study in 2021 found the MIR of SFTSV to be 0.01%, both in the city (two positive pools/16,803 ticks) and on pet trails (one positive pool/11,016 ticks), and it was notably higher at 0.29% (one positive pool/346 ticks) on pets. In contrast, no SFTSV was detected in pet-derived ticks in 2022. A previous survey in 2020 indicated an MIR of 0.2% for ticks, with a peak of 4.0% in some regions like An-dong (in the southern region of Korea) [31]. This variance in MIR across different areas highlights regional differences in tick-borne infection rates. Additionally, the MIR of ticks found on wild animals was 4.98%, significantly higher than what we observed for companion animals [32].

For *Anaplasma* spp., the MIR was 0.87% (three positive pools/346 ticks) in 2021 and 0.2% (two positive pools/1011 ticks) in 2022 among companion animals, with no detections in urban settings and along pet trails. This mirrors the findings from a Gyeongsang Province (in the southern region of Korea) survey in 2021. Following a survey of ticks with *Anaplasma* spp. infections in the area, the pathogen was confirmed in 27 pools among 3825 ticks, and the MIR was 0.7%. The lower MIR seen in 2022 might be attributed to a higher proportion of nymphs, which generally show a lesser disease prevalence [33]. This is similar to Daejeon’s 2021 pet infection rate. The reason for the relatively low MIR in 2022 may be related to the large proportion of nymphs collected compared to 2021. *Ehrlichia* spp. exhibited an MIR of 0.07% (eight positive pools/11,016 ticks) on pet trails and 0.87% (three positive pools/346 ticks) in pets in 2021, but it was not detected in 2022 on urban trails or pets. This suggests a higher relative infection rate in companion animals compared to their environment.

*Borrelia* spp. had an MIR of 0.19% (thirty-two positive pools/16,803 ticks) on urban trails, 0.03% (three positive pools/11,016 ticks) on pet trails, and 0.28% (one positive pool/346 ticks) from companion animals in 2021, with a slight increase to 0.1% (one positive pool/1011 ticks) from companion animals in 2022. A 2021 survey revealed a 0.33% MIR for *Borrelia* spp. from 20 pools of 6102 ticks. In particular, *I. nipponensis* had a very high infection rate, with an MIR of 34% (17 positive pools/50 ticks) [28]. In this study, *I. nipponensis* also showed a higher infection rate of 2.0% compared to other species. This seems to be because the main vector of *Borrelia* spp. is the genus *Ixodes*, including the deer tick (*Ixodes scapularis*) [34]. As a result, *Borrelia* spp. has been confirmed not only in the environment, but also in companion animals.

In this study, we discovered the possibility that tick distribution may have been affected by climate change in Korea. In addition, all four pathogens (SFTSV, *Anaplasma* spp., *Ehrlichia* spp., and *Borrelia* spp.) were confirmed to be present on companion animals and in the environment. This suggests the possibility that these four pathogens (SFTSV, *Anaplasma* spp., *Ehrlichia* spp., and *Borrelia* spp.) can be transmitted from companion animals to humans. Recently, there have been cases showing that, in addition to infections caused by ticks from these pets, direct infection can also occur. In the case of SFTSV, the common natural infection mechanism is known to be transmission through tick bites. However, the case of a woman in Japan, who was found to have SFTSV and died after being bitten by a stray cat that she was trying to rescue, was the first case of direct infection from an infected animal, and there was also a case where a veterinarian who was treating a cat was infected with SFTSV [35,36]. Therefore, it is difficult to dismiss the idea that other tick pathogens, such as *Anaplasma* spp., *Ehrlichia* spp., and *Borrelia* spp., may also be transmitted between animals or between animals and humans. Therefore, this possibility necessitates the ongoing monitoring of ticks on companion animals and in their surrounding environment. While our genus-level bacterial analysis has limitations, further sequencing research is necessary for a more comprehensive understanding.

## 5. Conclusions

In this study, we confirmed the distribution of ticks and four pathogens (SFTSV, *Anaplasma* spp., *Ehrlichia* spp., and *Borrelia* spp.) on trails and companion animals in Daejeon Metropolitan City, Korea. We collected ticks from trails and pet hospitals in the Daejeon area over a period of three years and confirmed the presence of pathogens through PCR analysis. As a result of examining approximately 29,200 ticks, the dominant species was found to be *H. longicornis*, and the presence of four types of pathogens was confirmed. These findings illuminate the possibility of tick-borne disease transmission between companion animals and humans and the emergence of a new tick (*A. testudinarium*) in the Daejeon area. The insights gained from this study are valuable for understanding the distribution of ticks and tick-borne pathogens in Korea and could inform the development of preventive measures against tick-borne disease transmission between companion animals and humans.

## Figures and Tables

**Figure 1 vetsci-11-00090-f001:**
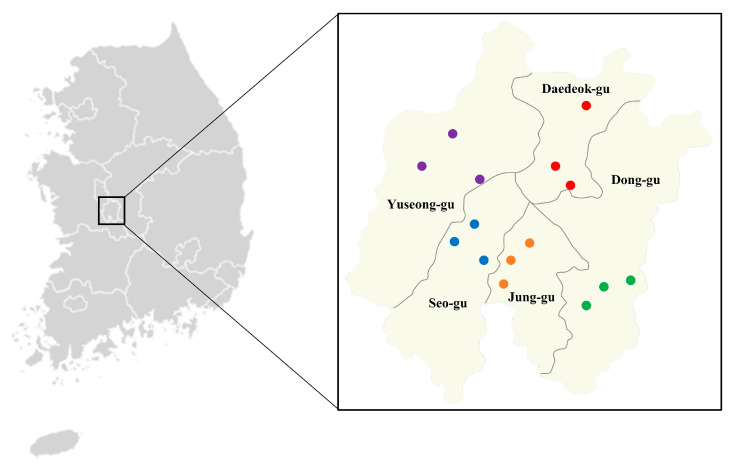
Collecting sites of ticks in Daejeon Metropolitan City, Korea. Daejeon Metroplitan City has five districts (Dong-gu, Jung-gu, Seo-gu, Yuseong-gu, and Daedeok-gu). The urban walking trails selected in each district are marked with dots (green, Dong-gu; orange, Jung-gu; blue, Seo-gu; purple, Yuseong-gu; red, Daedeok-gu).

**Table 1 vetsci-11-00090-t001:** Information regarding the urban walking trails in Daejeon Metropolitan City, Korea.

District	Administrative Location	GPS Location
Dong-gu	Gao Neighborhood park	36°30′63″ N 127°45′87″ E
	Choji park	36°27′85″ N 127°46′22″ E
	Sanseo-ro	36°28′35″ N 127°46′8″ E
Jung-gu	Sajeong park	36°29′92″ N 127°40′69″ E
	Daedunsan-ro	36°28′90″ N 127°37′23″ E
	Baekcheon Buddhist temple	36°29′92″ N 127°39′56″ E
Seo-gu	Sotae Neighborhood park	36°31′62″ N 127°34′37″ E
	Wolpyeong park	36°32′26″ N 127°36′02″ E
	Whaum Buddhist temple	36°29′90″ N 127°36′64″ E
Yuseong-gu	Sundusan Neighborhood park	36°37′25″ N 127°37′39″ E
	Sinsung Neighborhood park	36°38′5″ N 127°35′35″ E
	Bokyong Urban natural park	36°34′29″ N 127°53′18″ E
Daedeok-gu	Gilchi-Neighborhood park	36°36′19″ N 127°45′82″ E
	Birae Buddhist temple	36°37′9″ N 127°44′88″ E
	Jangdong Forest park	36°40′65″ N 127°43′88″ E

**Table 2 vetsci-11-00090-t002:** Information regarding the primers used for the detection of tick-borne pathogens.

Target Pathogen	Primer	Nucleotide Sequences (5′ → 3′)	Target Gene (Product Size)	Reference
SFTSV	MF3 MR2	GATGAGATGGTCCATGCTGATTCT CTCATGGGGTGGAATGTCCTCAC	M segment (560 bp)	[22]
	MMF3 MMF2	TAAACTTGTGTCGTGCAGGC CCCAGCGACATCTCCTTACA	M segment (245 bp)	
*Anaplasma* spp.	AE1-F1 AE1-R1	AAGCTTAACACATGCAAGTCGAA AGTCACTGACCCAACCTTAAATG	*16S* rRNA (1406 bp)	[23]
	AP-F AP-R	GTCGAACGGATTATTCTTTATAGCTTGC CCCTTCCGTTAAGAAGGATCTAATCTCC	*16S* rRNA (926 bp)	
*Ehrlichia* spp.	AE1-F1 AE1-R1	AAGCTTAACACATGCAAGTCGAA AGTCACTGACCCAACCTTAAATG	*16S* rRNA (1406 bp)	[23]
	EC-F2 EC-R2	CAATTGCTTATAACCTTTTGGTTATAAAT TATAGGTACCGTCATTATCTTCCCTAT	*16S* rRNA (390 bp)	
*Borrelia* spp.	B1 B2	TAGATGAGTCTGCGTCTTATTA CTTACACCAGGAATTCTAACTT	*16S* rRNA (465 bp)	[24]

**Table 3 vetsci-11-00090-t003:** The growth stages of and number of pathogens detected on tick species found on urban walking trails.

Species of Ticks	Stages	No. of Ticks	No. of Pools	No. of Detected Pathogens				Total
				SFTSV	*Anaplasma* spp.	*Ehrlichia* spp.	*Borrelia* spp.	
*H. longicornis*	Adult Nymph Larva Subtotal	1074 3952 9923 14,949	279 318 251 848	2 0 0 2	0 0 0 0	0 0 0 0	6 6 1 13	8 6 1 15
*H. flava*	Adult Nymph Larva Subtotal	76 224 711 1011	51 65 35 151	0 0 0 0	0 0 0 0	0 0 0 0	0 1 0 1	0 1 0 1
*I. nipponensis*	Adult Nymph Larva Subtotal	51 16 775 842	58 11 27 86	0 0 0 0	0 0 0 0	0 0 0 0	14 4 0 18	14 4 0 18
*A. testudinarium*	Adult Nymph Larva Subtotal	0 1 0 1	0 1 0 1	0 0 0 0	0 0 0 0	0 0 0 0	0 0 0 0	0 0 0 0
Total		16,803	1086	2	0	0	32	34
MIR (%)		-	-	0.01	0	0	0.19	-

**Table 4 vetsci-11-00090-t004:** The growth stages of and number of pathogens detected on tick species found on pet trails.

Species of Ticks	Stages	No. of Ticks	No. of Pools	No. of Detected Pathogens				Total
				SFTSV	*Anaplasma* spp.	*Ehrlichia* spp.	*Borrelia* spp.	
*H. longicornis*	Adult Nymph Larva Subtotal	1591 7232 1090 9913	186 212 23 421	1 0 0 1	0 0 0 0	3 2 0 5	0 0 0 0	4 2 0 6
*H. flava*	Adult Nymph Larva Subtotal	123 164 772 1059	51 26 21 98	0 0 0 0	0 0 0 0	1 1 0 2	0 0 0 0	1 1 0 2
*I. nipponensis*	Adult Nymph Larva Subtotal	33 10 0 43	25 4 0 29	0 0 0 0	0 0 0 0	1 0 0 1	3 0 0 3	4 0 0 4
*A. testudinarium*	Adult Nymph Larva Subtotal	0 1 0 1	0 1 0 1	0 0 0 0	0 0 0 0	0 0 0 0	0 0 0 0	0 0 0 0
Total		11,016	549	1	0	8	3	12
MIR (%)		-	-	0.01	0	0.07	0.03	-

**Table 5 vetsci-11-00090-t005:** The growth stage of and number of pathogens detected on tick species found on companion animals in 2021.

Species of Ticks	Stages	No. of Ticks	No. of Pools	No. of Detected Pathogens				Total
				SFTSV	*Anaplasma* spp.	*Ehrlichia* spp.	*Borrelia* spp.	
*H. longicornis*	Adult Nymph Larva Subtotal	275 35 0 310	93 11 0 104	1 0 0 1	1 0 0 1	2 0 0 2	1 0 0 1	5 0 0 5
*H. flava*	Adult Nymph Larva Subtotal	1 0 0 1	1 0 0 1	0 0 0 0	0 0 0 0	0 0 0 0	0 0 0 0	0 0 0 0
*I. nipponensis*	Adult Nymph Larva Subtotal	34 1 0 35	24 1 0 25	0 0 0 0	2 0 0 2	1 0 0 1	0 0 0 0	3 0 0 3
Total		346	130	1	3	3	1	8
MIR (%)		-	-	0.29	0.87	0.87	0.29	-

**Table 6 vetsci-11-00090-t006:** The growth stage of and number of pathogens detected on tick species found on companion animals in 2022.

Species of Ticks	Stages	No. of Ticks	No. of Pools	No. of Detected Pathogens				Total
				SFTSV	*Anaplasma* spp.	*Ehrlichia* spp.	*Borrelia* spp.	
*H. longicornis*	Adult Nymph Larva Subtotal	225 514 25 764	32 25 3 60	0 0 0 0	1 0 0 1	0 0 0 0	1 0 0 1	2 0 0 2
*H. flava*	Adult Nymph Larva Subtotal	66 74 0 140	10 5 0 15	0 0 0 0	0 0 0 0	0 0 0 0	0 0 0 0	0 0 0 0
*I. nipponensis*	Adult Nymph Larva Subtotal	107 0 0 107	18 0 0 18	0 0 0 0	1 0 0 1	0 0 0 0	0 0 0 0	1 0 0 1
Total		1011	93	0	2	0	1	3
MIR (%)		-	-	0	0.2	0	0.1	-

## Data Availability

The files containing the data supporting our findings can be requested directly to the corresponding author.

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
