# Peer review of "Tick Diversity and Pathogen Transmission in Daejeon, Korea: Implications from Companion Animals and Walking Trails"

_vetsci, 2024, doi:10.3390/vetsci11020090_

Round 1
Reviewer 1 Report
Comments and Suggestions for Authors
This paper provides an interesting overview of tick-borne diseases and tick distributions in different environments and among pets. This study effectively highlights the importance of surveying ticks and understanding disease distribution, particularly in light of climate change.
However, there are a few areas that could be improved upon. First, on line 142, needs to mention the concentrations of the extracted nucleic acid that were used. This information is important for helping readers fully understand the methods and results of the study. Additionally, in the material and methods section, it would be more appropriate to report nucleic acid concentrations used in PCR rather than "ul". This observation is applied to all the PCR-related sections in the paper.
Furthermore, on line 139, it is suggested to add a period after "A. testudinarium" and begin the next sentence with "A. longicorins appears to be…". This will help improve the clarity and flow of the sentence.
Moreover, on lines 245 and 246, the tick distribution is mentioned, but the table showing the distribution among the districts is missing. To enhance the reader's understanding, it is recommended to include a map displaying the districts. This approach will be particularly useful for readers who are unfamiliar with specific regions in Korea. Additionally, it is suggested to include a table in this section that presents the tick distribution and includes the minimum infection rate for each district and disease. This will provide a comprehensive overview of the data and make it easier for readers to interpret the results.
Another important aspect that could be addressed in the paper is the discussion on the tick distribution and the potential vector function of H. longicornis. While Borrelia was found in H. longicornis ticks, it would be valuable to discuss the implications of these findings and what experiments could be performed to verify whether H. longicornis can function as a vector for these diseases. Discussing the known vector functions and the ticks found with Anaplasma and Ehrlichia may also be beneficial.
Overall, the paper provides valuable insights into tick-borne diseases and tick distribution, but there are several areas where it could be improved. By addressing the aforementioned suggestions, the paper will be enhanced in terms of clarity, comprehensiveness, and relevance to readers.
Author Response
Critique #4. Another important aspect that could be addressed in the paper is the discussion on the tick distribution and the potential vector function of H. longicornis. While Borrelia was found in H. longicornis ticks, it would be valuable to discuss the implications of these findings and what experiments could be performed to verify whether H. longicornis can function as a vector for these diseases. Discussing the known vector functions and the ticks found with Anaplasma and Ehrlichia may also be beneficial.
Thank you for the reviewer’s comment. We added the following discussion of H. longicornis as a potential vector for pathogens. On line 284 ‘In addition to SFTVS, H. longicornis has been reported to have possibility as a vector that can transmit Borrelia spp., Anaplasma spp., and Ehrlichia spp. [27, 28, 29]. The detection of the four pathogens in our study may add weight to their possibility as potential vectors.’

Reviewer 2 Report
Comments and Suggestions for Authors
In this study Seo and colleagues assess the risk of tick-borne disease from companion animals in the Daejeon metropolitan area of Korea by collecting and PCR testing ticks from parks and walking trails used by companion animals, as well as directly from pets at vets and animals shelters. This combined approach leads to a good coverage of potential risk areas of the city, and leads to a large collection of ticks. This One Health approach is relatively novel in South Korea where there have been limited surveys of ticks and their pathogens in relation to companion animals.
While the tick collections provide some useful information on the most common tick species and lifestages encountered in these areas (and identify a new species in the area), the pathogen testing is less informative, and is the major weakness of this paper. Firstly, the testing strategy is not clearly described in the methods and results. Secondly, the bacterial pathogens identified in the study are only reported to genus level, which is uninformative because there are many species in these genera that are unclassified or not considered pathogenic. Thus, it is unclear why the authors report the use of nested PCR but do not sequence the resulting PCR products to identify pathogen species. qPCR assays are described in the methods but it is unclear why or how these were used. Ideally, these would be used to screen for presence of pathogens, which would then be confirmed in samples by nested PCR which could also identify pathogen species.
The authors also make some conclusions which are not supported by the data provided in their study. Overall the testing results need to be improved for this paper to provide a good assessment of which tick-borne bacteria are present and whether they might pose a risk to human and animal health.
Please find detailed comments below:
Abstract
line 28 - specify which pathogens were tested for.
line 29 - this should say 3 genera.
Introduction
The introduction is well written and gives a good background to the study.
line 46 - this list should also contain Borrelia, particularly as it is one of the bacteria under study here.
line 60-62: it seems like these two sentences have merged and are overlapping - it would be sufficient to only include the second part: "Similar patterns of tick migration and disease prevalence have been observed in European countries, including Scandinavia"
Materials & Methods
line 118 - what is special about these conical tubes?
line 138 & 140 - unnecessary to include "meticulously" here, this word can be deleted.
section 2.4 - from the presented results it is unclear if and at what point real-time PCR was used in the study.
One of the kits described also detects Rickettsia, so it would also be interesting to present findings for this group of bacteria which includes multiple important tick-borne pathogens.
It is unclear why nested PCR would be used in addition to real-time PCR if the products cannot delineate species, and are not being sequenced for pathogen species identification.
section 2.7 - the Borrelia PCR does not appear to be nested, as it contains a single pair of primers.
While the SFTSV detection is valid, there is less value in reporting bacterial pathogens to genus level as these genera contain multiple species that are non-pathogenic.
Results
Description of the results is somewhat unclear - instead of "samples" the authors should use "pools" or "pooled samples", and state "adult pools" or "nymph pools" etc.
MIR results should be given for each pathogen in the Results, rather than in the Discussion. MIR values should be described for each pathogen and tick species, and/or included in Tables 3-5.
Tables 3, 4, 5- SFTS should be changed to SFTSV.
Discussion
line 238: three genera
line 247, 249 and 250: SFTSV - the virus is being detected, not the disease.
line 261 - the reason for the decreased Anaplasma prevalence seen in 2022 could also be related to the fact that more nymphs were tested than in 2021, as nymphs usually show lower prevalences than adults.
line 280: this study did not confirm climate change is having an effect on ticks in the country, it only provides some support for this theory due to the finding of a tick species that is usually found in more southerly warmer areas.
line 283: it is also incorrect to say that this study "indirectly confirms that ticks from companion animals can increase the chance of infection". This study only showed that potentially pathogenic bacteria are present in ticks from companion animals, which could represent a health risk to their owners.
The title "5. Patents" should be deleted.
Comments on the Quality of English LanguageEnglish language is good throughout, but needs to be checked for spelling, as I noticed a few errors, for example:
line 19 and 242: H. longicornis
line 59: I. scapularis
line 172: Anaplasma
line 185: detection
Author Response
Critique #1. line 28 - specify which pathogens were tested for.
Thank you for the reviewer’s comment. We added information about the pathogens tested. On line 28, we changed that to ‘These ticks were then subjected to conventional (nested) PCR for SFTSV, Anaplasma spp., Ehrlichia spp., Borrelia spp.’
Critique #2. line 29 - this should say 3 genera.
Thank you for the reviewer’s comment. We apologize for the incorrect calculation of the genus and corrected it as follows. On line 29, We changed that to ‘We identified a total of 29,176 ticks consisting of 3 genera and 4 species’
Critique #3. line 46 - this list should also contain Borrelia, particularly as it is one of the bacteria under study here.
Thank you for the reviewer’s comment. We changed that to ‘Notably, genera such as Rickettsia, Anaplasma, Bartonella, Borrelia, Coxiella, and Ehrlichia are among the medically important pathogens transmitted by ticks’ on line 47.
Critique #4. line 60-62: it seems like these two sentences have merged and are overlapping - it would be sufficient to only include the second part: "Similar patterns of tick migration and disease prevalence have been observed in European countries, including Scandinavia"
Thank you for the reviewer’s comment. We modified it to remove the first sentence and use only the second sentence on line 61.
Critique #5. line 118 - what is special about these conical tubes?
Thank you for the reviewer’s comment. We apologize for providing incorrect information. This is not a specially manufactured tube, and we have described detailed tube information about it. On line 125 ‘tick collection tubes (SPL Life Sciences Co., Korea)’.
Critique #6. line 138 & 140 - unnecessary to include "meticulously" here, this word can be deleted.
Thank you for the reviewer’s comment. We deleted the word after confirming this.
Critique #7. section 2.4 - from the presented results it is unclear if and at what point real-time PCR was used in the study. One of the kits described also detects Rickettsia, so it would also be interesting to present findings for this group of bacteria which includes multiple important tick-borne pathogens.
Thank you for the reviewer’s comment. We first performed Real-time PCR and then additionally performed conventional PCR to detect pathogens more accurately. We added description for this on line 166.
Critique #8. section 2.7 - the Borrelia PCR does not appear to be nested, as it contains a single pair of primers.
Thank you for the reviewer’s comment. We re-checked the experimental process and confirmed that the experiment on Borrelia spp. was conducted only through real-time PCR, not conventional PCR. The detailed experimental process for this is described in Section 2-4 (line 171).
Critique #9. While the SFTSV detection is valid, there is less value in reporting bacterial pathogens to genus level as these genera contain multiple species that are non-pathogenic.
Thank you for the reviewer’s comment. We agree with the reviewer's opinion and added the limitations of our study to the discussion section (line 338).
Critique #10. Description of the results is somewhat unclear - instead of "samples" the authors should use "pools" or "pooled samples", and state "adult pools" or "nymph pools" etc.
Thank you for the reviewer’s comment. we reviewed the result and discussion sections and changed the descriptions.
Critique #11. MIR results should be given for each pathogen in the Results, rather than in the Discussion. MIR values should be described for each pathogen and tick species, and/or included in Tables 3-5.
Tables 3, 4, 5- SFTS should be changed to SFTSV.
Thank you for the reviewer’s comment. We added MIR for each pathogen in Table 3-6.
Critique #12. line 238: three genera
Thank you for the reviewer’s comment. We changed that to ‘we identified three genera and four species’ on line 278.
Critique #13. line 247, 249 and 250: SFTSV - the virus is being detected, not the disease.
Thank you for the reviewer’s comment. We reviewed the entire paper and changed SFTS to SFTSV where necessary.
Critique #14. line 261 - the reason for the decreased Anaplasma prevalence seen in 2022 could also be related to the fact that more nymphs were tested than in 2021, as nymphs usually show lower prevalences than adults.
Thank you for the reviewer’s comment. We agreed with the reviewer's opinion and added the description for this. On line 304, ‘The reason for the relatively low MIR in 2022 may be related to the large proportion of nymphs collected compared to 2021. In general, nymphs show a lower disease prevalence than adults’.
Critique #15. line 280: this study did not confirm climate change is having an effect on ticks in the country, it only provides some support for this theory due to the finding of a tick species that is usually found in more southerly warmer areas.
Thank you for the reviewer’s comment. We think that we used too strong an expression compared to our experiment results. We agree with the reviewer's opinion and changed the description of this. On line 324, ‘In this study, we discovered the possibility that tick distribution may have been affected by climate change in Korea’
Critique #16. line 283: it is also incorrect to say that this study "indirectly confirms that ticks from companion animals can increase the chance of infection". This study only showed that potentially pathogenic bacteria are present in ticks from companion animals, which could represent a health risk to their owners.
Thank you for the reviewer’s comment. We think that we used strong an expression compared to our experiment results. We agree with the reviewer's opinion and changed the description of this. On line 327, ‘This suggests the possibility that four pathogens (SFTSV, Anaplasma spp., Ehrlichia spp., Borrelia spp.) from companion animals can be transmitted to humans’
Critique #17. The title "5. Patents" should be deleted.
Thank you for the reviewer’s comment. We deleted that section.
Critique #17. line 19 and 242: H. longicornis.
Thank you for the reviewer’s comment. We corrected the typing error on line 19 and 282.
Critique #17. line 59: I. scapularis
Thank you for the reviewer’s comment. We corrected the typing error on line 59.
Critique #17. line 172: Anaplasma
Thank you for the reviewer’s comment. We corrected the typing error on line 197.
Critique #17. line 185: detection
Thank you for the reviewer’s comment. We corrected the typing error on line 219.
Reviewer 3 Report
Comments and Suggestions for Authors
Review Report of the manuscript ID: vetsci-2812924
Type of manuscript: Article
Title: Tick Diversity and Pathogen Transmission in Daejeon, Korea: Implications from Companion Animals and Walking Trails
Summary
This manuscript describes the collection and identification of ticks in the environment and on dogs and the detection of pathogens by molecular methods in the collected ticks. The study emphasizes the significance of climate changes. The strength of the study lies in the relevance of the topic; however, it has some weaknesses such as the lack of information on climatic data during the study period.
General comments
In general, the manuscript is confusing, poorly organized and some information is missing. The reference list is not extensive, with 12 out of 31 references published in the last 5 years. In my opinion, the publication of this study requires an extensive reorganization of the entire work, particularly in the Materials and Methods section. In this section, the authors should i) include the molecular identification of ticks, ii) explain how many ticks were included in each pool, iii) provide a clearer explanation of the molecular research on agents, especially the text between lines 149 and 153. The results section is extensive and confusing too.
Kind regards
Comments on the Quality of English LanguageAn extensive review is necessary.
Author Response
Critique 1. In general, the manuscript is confusing, poorly organized and some information is missing. The reference list is not extensive, with 12 out of 31 references published in the last 5 years. In my opinion, the publication of this study requires an extensive reorganization of the entire work, particularly in the Materials and Methods section. In this section, the authors should i) include the molecular identification of ticks, ii) explain how many ticks were included in each pool, iii) provide a clearer explanation of the molecular research on agents, especially the text between lines 149 and 153. The results section is extensive and confusing too.
Thank you for the reviewer’s comment. We added four more references from the last five years to the discussion section. About materials and methods section, we added molecular identification of ticks on line 135 and described results on line 212. Additionally, we described how many ticks were pooled on line 154 and provided a more detailed description of the experiment in the PCR section.

Round 2
Reviewer 2 Report
Comments and Suggestions for Authors
In this revision, the authors have addressed my major concerns with the study - the methods and results sections have been improved and are now much clearer, and the limitation of identifying bacteria to genus level has been included in the Discussion. I also like the map figure that has been added.
In the conclusions section, I would recommend that authors delete the words "due to global warming" from line 350, as this was not proven by this study. I look forward to future studies on the expansion of A. testudinarium in Korea!
Comments on the Quality of English LanguageI noticed the following spelling mistakes:
line 76: Among these,
Throughout Section 2.1 Daejeon is written as Dajeon.
line 171: "...were used according to the manufacturer’s instructions."
line 216: correct 'specie' to 'species'
line 232 & 247: correct spelling of "detection" in headings
Author Response
We are pleased to resubmit a revised manuscript entitled “Tick Diversity and Pathogen Transmission in Daejeon, Korea: Implications from Companion Animals and Walking Trails” for reconsideration in Veterinary sciences as an original manuscript. We have carefully evaluated the reviewer’s comments and have provided a point-by-point response below. Changes in the manuscript have been identified by page and sentence, and note by blue FONT. We hope that the revised manuscript meets the reviewers’ expectations at Veterinary sciences.
Critique #1. In the conclusions section, I would recommend that authors delete the words "due to global warming" from line 350, as this was not proven by this study.
Thank you for the reviewer’s comment. We agree with the reviewer's opinion and delete the description of this. On line 353, ‘These findings illuminate the possibility of tick-borne disease transmission between companion animals and humans and the emergence of a new tick (A. testudinarium) in Daejeon area’
Critique #2. line 76: Among these
Thank you for the reviewer’s comment. We corrected the typing error on line 76.
Critique #3. Throughout Section 2.1 Daejeon is written as Dajeon.
Thank you for the reviewer’s comment. We corrected the typing error on Section 2.1.
Critique 4. line 171: "...were used according to the manufacturer’s instructions."
Thank you for the reviewer’s comment. It was revised on line 171.
Critique 5. line 216: correct 'specie' to 'species'
Thank you for the reviewer’s comment. We corrected the typing error on line 221.
Critique 6. line 232 & 247: correct spelling of "detection" in headings
Thank you for the reviewer’s comment. We corrected the typing error on line 237 and 252.

Reviewer 3 Report
Comments and Suggestions for Authors
Review Report of the manuscript ID: vetsci-2812924_R1
Type of manuscript: Article
Title: Tick Diversity and Pathogen Transmission in Daejeon, Korea: Implications from Companion Animals and Walking Trails
Summary
This manuscript describes the collection and identification of ticks in the environment and on dogs and the detection of pathogens by molecular methods in the collected ticks. The study emphasizes the significance of climate changes. The strength of the study lies in the relevance of the topic; however, it has some weaknesses such as the lack of information on climatic data during the study period.
General comments
In general, the manuscript has been improved particularly regarding the organizations of information. However, some mentioned points raise some questions such as:
- the molecular identification of ticks was conducted previously but not described? Why?
- I did not find in the MS a reference to the meaning of MIR(%)
- I don't see in the MS any evidence that supports the statement in line 350 "the emergence of a new tick (A. testudinarium) due to global warming in Daejeon area"
Comments on the Quality of English LanguageThe manuscript needs revision by a native English speaker.
Author Response
Critique #1. the molecular identification of ticks was conducted previously but not described? Why?
Thank you for the comment. Actually, we were planning to write a separate paper on the content, and there was advice from a tick specialist that classification of ticks is sufficient with visual analysis.
Critique #2. I did not find in the MS a reference to the meaning of MIR (%).
Thank you for the reviewer’s comment. We added description about Minimum infection rate (MIR) on line 210 and reference of that description [26].
Critique #3. I don't see in the MS any evidence that supports the statement in line 350 "the emergence of a new tick (A. testudinarium) due to global warming in Daejeon area"
Thank you for the reviewer’s comment. We agree with the reviewer's opinion and delete the description of this. On line 352, ‘These findings illuminate the possibility of tick-borne disease transmission between companion animals and humans and the emergence of a new tick (A. testudinarium) in Daejeon area’
